# VOILA-A: ALIGNING VISION-LANGUAGE MODELS WITH USER'S GAZE ATTENTION

## ABSTRACT

In recent years, the integration of vision and language understanding has led to significant advancements in artificial intelligence, particularly through Vision-Language Models (VLMs). However, existing VLMs face challenges in handling real-world applications with complex scenes and multiple objects, as well as aligning their focus with the diverse attention patterns of human users. In this paper, we introduce gaze information, feasibly collected by AR or VR devices, as a proxy for human attention to guide VLMs and propose a novel approach, Voila-A, for gaze alignment to enhance the interpretability and effectiveness of these models in real-world applications. First, we collect hundreds of minutes of gaze data to demonstrate that we can mimic human gaze modalities using localized narratives. We then design an automatic data annotation pipeline utilizing GPT-4 to generate the VOILA-COCO dataset. Additionally, we innovate the Voila Perceiver modules to integrate gaze information into VLMs while preserving their pretrained knowledge. We evaluate Voila-A using a hold-out validation set and a newly collected VOILA-GAZE Testset, which features real-life scenarios captured with a gaze-tracking device. Our experimental results demonstrate that Voila-A significantly outperforms several baseline models. By aligning model attention with human gaze patterns, Voila-A paves the way for more intuitive, user-centric VLMs and fosters engaging human-AI interaction across a wide range of applications.

## 1 INTRODUCTION

The integration of vision and language understanding has witnessed significant advancements in recent years, particularly through the development of Vision-Language Models (VLMs). These models have demonstrated remarkable performance in various tasks, such as visual question answering, image captioning, and visual storytelling, among others. However, despite their impressive performance, current VLM systems often lack the ability to align their focus with that of a human user, which can lead to suboptimal performance and reduced user satisfaction. To address this challenge, we introduce Voila-A, a novel approach for aligning VLMs with a user's gaze attention, aiming to improve the interpretability and effectiveness of these models in real-world applications.

Recent research in multimodal vision and language tasks has leveraged multimodal large language models (MLLMs) to achieve superior performance (Liu et al., 2023; Li et al., 2023b; Alayrac et al., 2022). These models primarily focus on learning alignment between vision input and text tokens for LLMs or designing learnable interaction layers to attend the vision input to the frozen LLM layers. The importance of aligning AI with human attention has been highlighted in previous research, which demonstrates that incorporating visual attention can lead to improved user experience (Land, 2006; Tanriverdi & Jacob, 2000; Piening et al., 2021). Additionally, there has been growing interest in grounded MLLMs, which investigate the fine-grain grounding capability between region-text pairs instead of image-text pairs and further conduct dense regional prediction tasks (Zhong et al., 2022; Jin et al., 2023; Zhou et al., 2023).

Visual regions can be represented in various ways, such as bounding boxes (Zhu et al., 2016; Liu et al., 2017), points (Mani et al., 2020), or traces (Pont-Tuset et al., 2020; Yan et al., 2021). To input regional information into models, several methods have been proposed, including concatenating cropped image patches with the original text/image as model input (Zhang et al., 2023b; Bracha et al., 2023), using masks or Gaussian maps to emphasize areas of user interest Lin et al.

(2020; 2022), or encoding points, boxes, or traces into positional encodings (Kirillov et al., 2023; Voigtlaender et al., 2023). While bounding boxes and points have been widely used in VLMs, gaze offers a more natural way to represent visual regions and are most similar to human gaze. In this work, we propose using gaze as a more convenient and interactive way of representing visual regions, especially in augmented reality and virtual reality scenarios. Specifically for gaze, previous works have proposed gaze-directed visual grounding (Qian et al., 2023) and eye-gaze-guided vision transformers (Ma et al., 2023). However, these approaches have limitations in terms of scalability and flexibility. Despite these advancements, the integration of gaze information into large VLMs remains an open challenge. A key challenge in achieving this alignment lies in the integration of gaze information into VLMs while preserving the pretrained knowledge.

To tackle this issue, we demonstrate that mouse trace data can be a proxy for gaze behavior modeling and leverage trace data from Localized Narratives (Pont-Tuset et al., 2020) to annotate instructional data using GPT-4 (OpenAI, 2023). We further design Voila-A's attention mechanism to incorporate gaze information while not forgetting pretrained knowledge. We evaluate Voila-A through a hold-out validation set and a newly collected test set of 100 samples, featuring real-life scenarios with a gaze-tracking device.

In this paper, we make the following contributions:

- We propose Voila-A, a novel approach for aligning VLMs with a user's gaze attention, and design innovative mechanisms to integrate gaze information into VLMs while preserving pretrained knowledge.

- We leverage trace data from Localized Narratives to annotate instructional data using GPT-4, generating the VOILA-COCO dataset with 72k QA pairs, and demonstrate the scalability of this method.

- We evaluate Voila-A through a hold-out validation set and a newly collected VOILA-GAZE test-set of 100 real gaze samples, demonstrating that our approach significantly outperforms several baselines, including Otter (Li et al., 2023a) and Kosmos-2 (Peng et al., 2023).

Voila-A represents a significant step forward in the development of more intuitive and user-centric VLMs. By aligning model attention with human gaze patterns and leveraging state-of-the-art techniques, we pave the way for more effective and engaging human-AI interaction in a wide range of applications.

## 2 BRIDGING THE GAP IN DAILY LIFE USAGE OF CURRENT VLMS THROUGH GAZE INTEGRATION

Although Vision-Language Models (VLMs) exhibit strong performance in various tasks, their applicability in everyday scenarios is hindered by their limited alignment with human users' focus. This misalignment leads to suboptimal performance and decreased user satisfaction. Current VLMs' inability to process these intentional modalities results in imprecise and unhelpful responses, as demonstrated in Figure 1. As shown in Figure 7, a user's intent can be communicated through spoken language, multimodal expressions, or even be concealed. Gaze direction data can clarify vague expressions, while uncovering hidden intentions is more challenging.

Zhang et al. (2020) provides an overview of gaze-related research, outlining a process that begins with collecting human gaze data (further discussed in G.3), followed by building models to predict human attention distribution (i.e., saliency models, as discussed in G.4), and culminating in human-gaze-assisted AI. They acknowledge that *AI agents capable of perceiving and understanding human gaze behaviors can better infer user needs and assist in daily tasks*. However, they also note that *research in this final direction is still limited*. Our work aims to advance this area further. Incorporating gaze data into VLMs improves their applicability and effectiveness in everyday scenarios. This integration enables VLMs to focus on users' interests, delivering accurate and relevant responses while understanding intent, context, and preferences. As spatial computing advances, gaze data becomes essential for dynamic, interactive environments, allowing VLMs to adapt in real-time and offer intuitive, seamless user experiences.

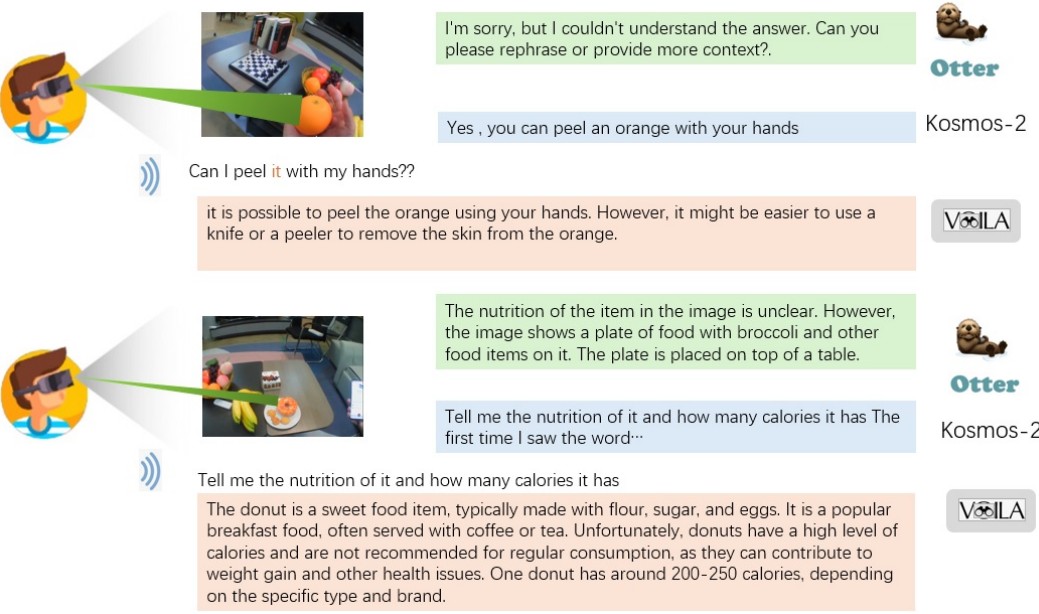

Figure 1: AR and VR scenarios usually involve complex scenes with multiple objects. Users may interested in only one specific object and gaze is the most natural way to interact with the device.

# 3 LEVERAGING TRACE DATA AS AN ALTERNATIVE APPROACH TO ALIGN VLMS WITH GAZE ATTENTION

Obtaining gaze data for training VLMs can be challenging, as it is difficult to annotate and expensive to acquire. However, an alternative approach can be employed to align VLMs with user gaze attention: utilizing trace data, such as mouse traces, which we demonstrate to have similarities with gaze data. In this section, we discuss the potential of trace data as a proxy for gaze data and propose a method for transforming trace data to make it more gaze-like, ultimately enabling the effective use of trace data for aligning VLMs with user gaze attention.

Localized Narratives (Pont-Tuset et al., 2020), a prior work, have annotated 849,000 images with mouse traces that are aligned with each word of the descriptions. The project involved 156 professional annotators who worked full-time, with annotator managers ensuring high-quality annotations through manual inspections and an automatic quality control mechanism. After discarding 23.5% of annotations, the remaining ones demonstrated a semantic accuracy of 98.0% for nouns and verbs. The accuracy of mouse traces in relation to object locations was also analyzed, revealing that most trace points were within the correct bounding box.

By collecting hundreds of minutes of gaze data samples as described in B, we find that gaze and mouse traces exhibit similarities, as users tend to fix their gaze on the target object when asking questions, a behavior also observed with mouse traces. However, there are minor differences between the two, specifically in terms of gaze fixation continuity and the presence of noise points outside the target object at the end of a query. In the case of mouse traces, points that fell outside the bounding box were attributed to two factors: annotators often circled around objects, causing the traces to be near but not inside the box, and some annotators started moving the mouse before describing the object or vice versa. These observations provide valuable insights for properly leveraging trace data into the alignment process and understanding the relationship between gaze attention and language description.

In order to utilize trace data as a substitute for gaze data, we introduce a method to transform mouse traces, thereby reducing the discrepancies between the two data types and making the trace data more gaze-like. We first address the inherent noise in both trace points and gaze points by converting them into 2D heatmaps using Gaussian blur:

$$H(x, y) = \frac{1}{2\pi\sigma^2} e^{-\frac{x^2 + y^2}{2\sigma^2}} \tag{1}$$

where $H(x, y)$ represents the heatmap value at position $(x, y)$, and $\sigma$ is the standard deviation of the Gaussian kernel.

Since mouse traces are more continuous than gaze fixations, we downsample the trace data to better resemble the distribution of gaze data. We investigate the Earth Mover's Distance (EMD) between the mean heatmaps of 1k gaze and trace samples while varying the sampling rate from 1 to 40:

$$\text{EMD}(P, Q) = \frac{\sum_{i=1}^{n} |F_i(P) - F_i(Q)|}{\sum_{i=1}^{n} F_i(P)} \tag{2}$$

where $P$ and $Q$ are the distributions of the gaze and trace heatmaps, $F_i$ denotes the cumulative distribution function, and $n$ is the number of bins.

We observe that the EMD has a local minimum value around a sampling rate of 25 as shown in Figure 2. By selecting this optimal sampling rate, we can approximate the trace heatmap as an alternative to the real gaze heatmap from a statistical perspective. Consequently, this transformation mitigates the differences in inter-relationships, compactness, and noisiness between the trace and gaze data.

## 4 METHOD

### 4.1 AUTOMATIC DATA ANNOTATION FOR LN-COCO

The automatic data annotation process for Voila-A is driven by the motivation to develop a more intuitive and user-centric VLM by aligning model attention with human gaze patterns. As shown in Figure 3, this process aims to create an effective and engaging human-AI interaction experience across various applications. To achieve this, we have designed an innovative prompting approach that leverages the capabilities of GPT-4 as a visual assistant to annotate trace-aligned instructional data to simulate the user's gaze attention. The data annotation process follows design principles to ensure accu-

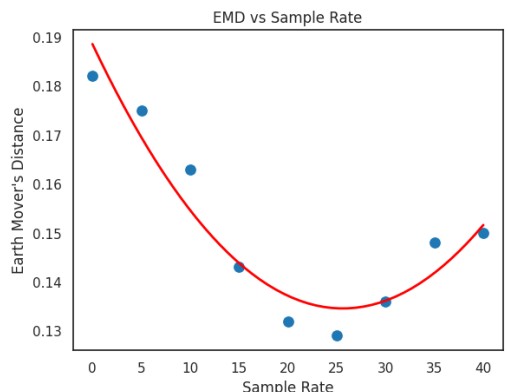

Figure 2: EMD between the mean heatmaps of 1k gaze and trace samples with varying sampling rates.

rate, relevant, and consistent annotations. These include: 1) focusing on referable sentences and appropriate tags, 2) using a conversational format with specific and general questions, 3) addressing various visual content aspects with definite answers, 4) incorporating complex questions while avoiding uncertainty, and 5) offering detailed, well-organized explanations.

As illustrated in Figure 3, the automatic data annotation pipeline comprises three stages.

**Stage 1: Prompt Design Iteration.** The first stage focuses on refining the prompt design. Let $S = \{(I_i, N_i, T_i, C_i)\}_{i=1}^{100}$ be a set of 100 samples from the LN-COCO dataset, where $I_i$ represents the image, $N_i$ the localized narrative, $T_i$ the corresponding trace, and $C_i$ the set of five captions from COCO-caption. We initiate the process with a basic system prompt, instructing GPT-4 to generate direct questions $Q_{i,j}^D$ and indirect questions $Q_{i,j}^I$ and corresponding answers $A_{i,j}$ that specifically reference the localized narratives while considering COCO-caption as background information. The referring portions are annotated with a unique marker $\mathcal{M}$ for trace matching during

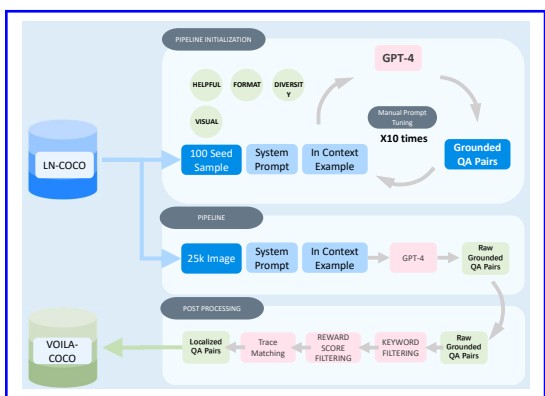

Figure 3: Automatic Data Annotation Pipeline

Table 1: Statistics of Voila-COCO and Voila-Gaze Datasets

| Dataset | Split | Number of Images | Number of Questions | Survival Rate from Raw Data |
|---|---|---|---|---|
| Voila-COCO | Training | 20000 | 70000 | 93.5% |
| Voila-COCO | Validation | 100 | 550 | 71.1% |
| Voila-COCO | Test | 500 | 1900 | 75.7% |
| Voila-GAZE | Real-life | 100 | 100 | 22.2% |

post-processing. We also provide two in-context examples to guide the model in generating helpful, well-formatted, diverse, and visually grounded QA pairs. Throughout each iteration $k$, we manually evaluate the quality of the generated grounded QA pairs and adjust the prompt to enhance their helpfulness, formatting, diversity, and visual relevance. After $K = 10$ iterations, we find the quality of most pairs to be satisfactory, and subsequently, we freeze the prompt to initiate the pipeline.

**Stage 2: Data Sampling.** In the second stage, we sample $N = 25,000$ image pairs from the LN-COCO dataset and obtain approximately $M = 75,000$ QA pairs.

**Stage 3: Post-processing.** The third stage involves post-processing the raw grounded QA pairs. This includes further filtering based on a set of keywords $\mathcal{K} = \{"prompt", "this picture", "reference caption", ...\}$. We define a filtering function $F_k(Q_{i,j}, A_{i,j}, \mathcal{K})$ that identifies and removes QA pairs containing meta descriptions of the prompt. We note that this issue may be further resolved by using GPT-4V, which was not available during our submission date. Additionally, we identify cases where answers are unhelpful, such as "I don't know" or "It's hard to tell." We find that these types of answers have low reward scores, so we further examine all pairs using a reward model (rew, 2023) and filter the dataset by setting a minimum reward threshold $\tau$. We define a filtering function $F_r(Q_{i,j}, A_{i,j}, \tau)$ that removes QA pairs with reward scores below $\tau$. Finally, we segment each localized narrative into temporally aligned segments with respect to the special marker $\mathcal{M}$. Each segment comprises a grounded fact, a corresponding trace, a direct and indirect question, and an answer. This forms the final VOILA-COCO dataset, denoted as $\mathcal{D} = \{(F_i, T_i, Q_{i,j}^D, Q_{i,j}^I, A_{i,j})\}$. It is worth noting that we did not utilize all localized narratives, leaving room for future exploration. We annotate the COCO subset of localized narratives, resulting in the Voila-COCO dataset, with statistics presented in Table 1.

The finalized prompt can be found in C. We also visualize a sample of our annotated data in Figure 11. By adhering to these design principles, the automatic data annotation process ensures that the resulting dataset is of high quality and effectively aligns the VLM's attention with that of a human user.

## 4.2 VOILA-GAZE: REAL-LIFE GAZE-QA PAIRS

To further demonstrate the effectiveness of our method in aligning VLMs with real-life users' gaze attention, we conduct experiments in two everyday scenarios, encompassing a variety of question types details can be found in Table 5.

In addition to the recorded gaze trajectory, video, and transcription, each participant is instructed to annotate the key elements of their questions, formulate clear questions based on their interpretations, and choose the best answer from three candidate answers generated by GPT-4 according to their annotations. The experiment includes 16 participants (8 per scenario) with an equal gender distribution, aged between 20 and 30 (with a standard deviation of 2.06). Each participant takes approximately 240 minutes to complete the study. After applying post-filtering and manual checking, we curate a set of 100 QA pairs as our real-life benchmark, VOILA-GAZE.The curation process is conducted by two individuals sequentially, with the second person double-checking the following aspects: **1. The question is related and aligned with gaze. 2. The answer is meaningful and can be considered a proper response to the gaze-directed question. 3. The question is not related to specific brands, prices, or any other objects beyond general knowledge. 4. The question type is not biased towards a few simple patterns.** This two-step process ensures the quality and relevance of the curated data while minimizing potential biases and maintaining a focus on general knowledge. Samples of VOILA-GAZE are shown in Figure 9.

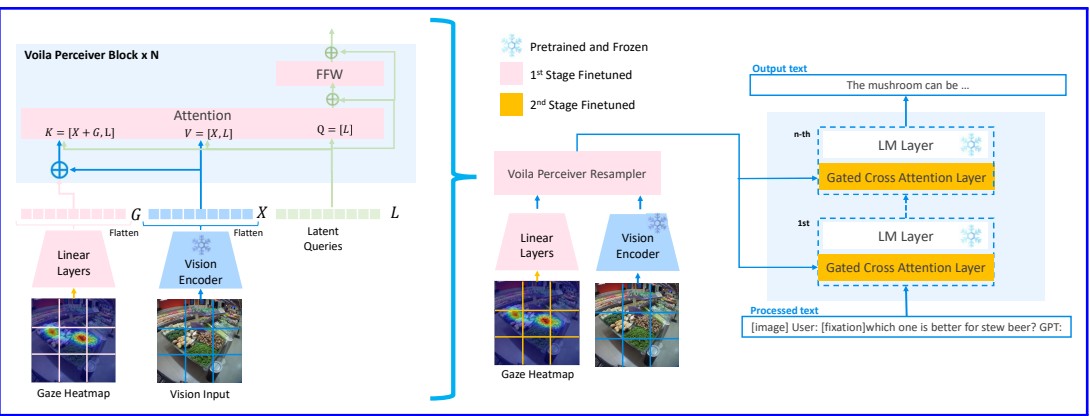

Figure 4: Overall Model Structure

## 4.3 MODEL DESIGN

We employ the model architecture from OpenFlamingo, as illustrated on the right side of Figure 4. This framework consists of a pre-trained vision encoder, language decoder, and gated cross-attention layers, offering flexibility for multi-image and multi-turn conversational interactions. The primary challenge lies in incorporating gaze instructional signals into a pre-trained VLM. To tackle this issue, we initially develop several potential solutions, which are discussed in Sec 5.3.2. Based on empirical evidence, we ultimately confirm the effectiveness of the Voila Perceiver Resampler solution. This approach comprises a series of Voila Perceiver Blocks (depicted on the left side of Figure 4). This innovative attention mechanism leverages gaze information to enhance visual feature perception. Our design adheres to the principle that gaze serves as an information aggregator in the attention process without disrupting the original learned distribution. The Voila Perceiver Block is defined as follows:

$$\textbf{VoilaPerceiverBlock}(x, L, G) = \text{LN}(L + \text{FF}(L + \text{Attn}(x, L, G))) \tag{3}$$

where $x$ represents the image features, $G$ is the gaze heatmap patch encoded after simple linear layer. $L$ denotes the latent features. It is introduced from the original Perceiver as a small set of latent units that forms an attention bottleneck through which the inputs must pass. The attention mechanism, $\text{Attn}(x, L, G)$, is computed as follows:

$$Q = \text{Linear}(L), \quad K = \text{Linear}(x \oplus L) + \text{Linear}(G \oplus P),$$
$$V = \text{Linear}(x \oplus L), \quad \text{Attn}(x, L, G) = \text{Softmax}(QK^\top)V \tag{4}$$

Here, $\oplus$ denotes concatenation along the feature dimension, $P$ is zero padding with the same shape as $L$ and $\text{Linear}(\cdot)$ represents a linear transformation. The feed-forward network, $\text{FF}(\cdot)$, is a sequence of layer normalization, linear transformation, GELU activation, and another linear transformation.

The Voila Perceiver Resampler processes the input image features and gaze information, and then feeding them into series of Voila Perceiver Block:

$$\textbf{VoilaPerceiverResampler}(X, G) = \text{LN}(\text{Layer}(X, L, G)) \tag{5}$$

where $X$ denotes the input image features, $G$ represents the gaze information, and $\text{Layer}(X, L, G)$ is a sequence of Voila Perceiver Blocks. To obtain the gaze information $G$, we first divide the gaze heatmap $G'$ into patches. Then, we apply a linear transformation followed by layer normalization. The process can be represented by the following equation:

$$G = \text{LN}(\text{Linear}(G')) \tag{6}$$

## 4.4 TRAINING

Our approach utilizes the OpenFlamingo training paradigm to train the Voila model, building upon the pretrained weights of the Otter model, which incorporates an MPT-7B(Team, 2023) language

encoder and a CLIP ViT-L/14 (Radford et al., 2021) vision encoder. To avoid overfitting and maximize the benefits of pretrained knowledge, we initially freeze both encoders. As shown in Figure 4, we then train only the linear layers directly related to gaze input at the first stage for one epoch before fine-tuning the entire Perceiver resampler module, the cross-attention layers integrated into the language encoder, and the input/output embeddings of the language encoder in second stage for an additional epoch. This process results in roughly 1.3 billion trainable parameters for the Otter model.

During training, we adhere to a specific format for preparing our training data. This format combines an image, user instruction, "GPT"-generated answers 1, and a unique token known as the [endofchunk] token. We arrange the training data as follows:

<context> [image] User:[fixation]<instruction> GPT:[answers] <answer>.[endofchunk]

Here, the [image], [answer], [fixation], and [endofchunk] tokens are distinct and serve particular functions. We adopt a chatbot-like format to enhance the instruction-following capabilities and conversational generalizability of our model. The [image] and [endofchunk] tokens originate from the OpenFlamingo training paradigm, while the [answer] token is a new addition by Otter. The [answer] token separates answers from instructions, allowing us to mask all tokens following the [answer] token during training and designate them as the model's prediction objectives. We also introduce the [fixation] token to direct the language model to utilize gaze information. We train our model using a cross-entropy loss function.

## 5 EXPERIMENT

### 5.1 EVALUATION METRICS

**GPT-4 RANKING**   We utilize GPT-4 RANKING as our primary automated evaluation metric to assess model performance through a one-to-one comparison. The GPT Ranking represents the language model's evaluation of the quality of the generated response. This score signifies the extent to which the response aligns with the ground truth image description and answer while demonstrating the model's language proficiency. Factors such as grammar, semantics, and fluency are taken into account when comparing the response to that of another model. **It is important to note that GPT-4 exhibits sequence ordering bias.** To mitigate this issue, we implement a dual-setting approach that reverses the order of the models, ensuring that the order does not influence the outcome. The prompt and evaluation procedure can be found in Figure 12.

**Reward Score**   Given that our dataset is automatically annotated using GPT-4, it is crucial to mitigate any potential system bias during model evaluation. To this end, we incorporate human preference by utilizing a reward model score as an auxiliary metric. The reward model, which assesses the human-like quality of a response, is trained using human feedback to predict the superiority of a generated answer in relation to a given question from a human perspective(rew, 2023). This approach allows us to achieve a more balanced and robust evaluation process, ensuring that our model's performance aligns with human expectations and preferences.

### 5.2 MAIN RESULTS

#### 5.2.1 VOILA EXHIBITS A BALANCED CAPABILITY BETWEEN HELPFULNESS AND FACT GROUNDING

In Figure 5, we observe a notable superiority of VOILA over both Otter and Kosmos-2 on the VOILA-COCO-TESTSET. Regarding the *grounding* capability, both VOILA and Kosmos-2 trained with fine-grained grounded facts outperform Otter model in a large extent. Besides, VOILA surpasses Kosmos-2 marginally. With respect to *helpful*

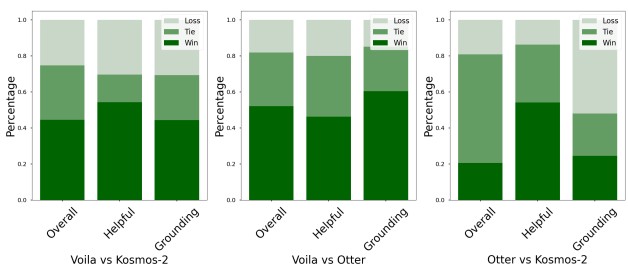

Figure 5: GPT-RANKING ON VOILA-COCO-Testset

capability, Otter delivers significantly more helpful responses than Kosmos-2. Since Otter is trained on top of Openflamingo with instruction-following dataset, it can provide more helpful response especially for informative queries while Kosmos-2 tends to answer visual observation from the input image. In addition, VOILA trained on gaze dataset demonstrates stronger helpful capabilities over all models.

In real gaze scenarios, as illustrated in Figure 6, VOILA outperforms the two baseline models as well. These scenarios differ substantially from the collected COCO images and present more challenging questions, necessitating a higher degree of accurate intent understanding and reasoning. Especially from the comparison of Otter vs Kosmos-2, we found that there are much more Tie results due to the hardness of the real cases. Despite these increased demands, VOILA continues to surpass both models, further showcasing its balanced proficiency in both helpfulness and fact grounding.

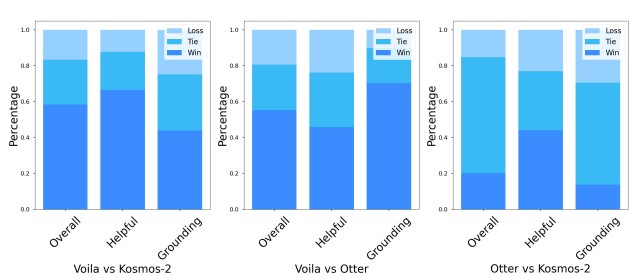

Figure 6: GPT-RANKING ON VOILA-GAZE

## 5.3 ABLATION STUDIES

### 5.3.1 QUERY TYPES HAS A SIGNIFICANT IMPACT ON RESPONSE QUALITY

Table 2: Ablation on query types

| Methods | Question types | Winning Rate Over Otter-base | Reward Score |
|---------|----------------|------------------------------|--------------|
| Otter-base | coreference query | - | -1.91 |
| Otter-base | direct query | 0.51 | 0.02 |
| Voila | coreference query | 0.41 | -0.79 |
| Voila | direct query | 0.62 | 0.14 |
| Voila | in-context prompt + coreference query | 0.46 | -0.02 |
| Voila | in-context prompt + direct query | 0.77 | 0.20 |

Table 2 investigates the varying performance of different question types, specifically direct and implicit/coreference queries. As the base model Openflamingo was pretrained on direct queries, both Otter and VOILA perform better in this category as expected. In addition, it is nature for human to communicate with correfernce queries. VOILA maintains strong performance when handling coreference queries with the gaze as a guidance while the Otter model decreases a lot. Furthermore, we append in-context QA pairs prior to the current query and observe that the examples further improves the quality of the responses. In real-life situations, multi-turn conversations are involved in most interactions with many coreference queries. The design of in-context prompt can assist VOILA to demonstrate a superior ability. This improvement is evident across both direct and coreference query types.

### 5.3.2 HEATMAP IS A BETTER WAY TO INCORPORATE GAZE

Table 3: Ablation on Methods of Integrating Gaze Data

| Methods | Wining Rate over Otter-base | Reward Score |
|---------|------------------------------|--------------|
| Otter-base | - | -1.91 |
| Gaze as discrete position tokens | 0.19 | -2.44 |
| Gaze *bounding box* as image patch | 0.36 | -1.26 |
| Gaze *bounding box* as discrete position tokens | 0.21 | -1.72 |
| Voila(Gaze as heatmap) | 0.41 | -0.79 |

To establish the effectiveness of our approach, we implemented several alternative methods for incorporating gaze data into VLMs. These methods include: converting gaze sequences into discrete

position tokens for LLMs, using the bounding box position of trace trajectories as additional patch tokens concatenated to VIT image feature token lists, and converting the bounding box coordinates into discrete tokens. We provide an illustration of those method in Figure 13. However, all these methods failed to outperform the gaze heatmap approach, as shown in Table 3.

### 5.3.3 GRADUAL UNFREEZING OF PARAMETERS YIELDS BETTER RESULTS

Table 4: Ablation on Training Procedure

| Layers fine-tuned | Winning Rate Over Otter-base | Reward Score |
|---|---|---|
| Otter-base frozen all | - | -1.91 |
| Otter-base vision perceiver+cross attention | 0.25 | -1.78 |
| Voila gaze weight | 0.24 | -1.52 |
| Voila gaze weight+LORA | 0.23 | -1.02 |
| Voila gaze weight→perciever+cross attention | 0.41 | -0.79 |

Table 4 presents empirical findings that demonstrate the effectiveness of gradually unfreezing model parameters. Instead of directly unfreezing the vision perceiver and cross-attention layers, or using LORA to fine-tune the entire model, we first fine-tune the gaze-related weights and then fine-tune the other parts of the perceiver and cross-attention layers, which yields better results. We hypothesize that this improvement is due to the newly added gaze component needing to adapt to the distribution of the pretrained layers first. This adaptation process can further help mitigate the issue of catastrophic forgetting.

### 5.4 QUALITATIVE STUDIES

We conduct qualitative studies on randomly selected cases and demonstrate the results of several representative examples in Appendix Figure 8. According to the analysis, the conclusions can be summarized as: 1) Existing models are able to generate reasonable results for **explicit** queries. In the 1st row, the object *cakes* and the attributes *color* are explicitly mentioned in the query, and the three models are able to answer (partially) correctly; 2) Regarding to **coreference** queries, the model Otter is hard to understand the pronouns like *it* without spatial guidance as shown in the 2nd row. This requires further context or generates the answer based on the salient object like *plane* instead of the actual human attention; 3) The Kosmos-2 model can take the **bounding box** for grounding as spatial guidance, it is sometimes not accurate compared to the heatmap used in VOILA. As shown in the 3rd row, the bounding box is too coarse and made the model focus on the object *plane* instead of the actual human attention *sky*; 4) Besides, we found that Kosmos-2 tends to describe the detailed visual content and sometimes lacks the **instruction-following** capability; In the 4th row, the Kosmos-2 responses *Keyboard* depicated in the bounding box ignoring the actual query intention; Finally, 5) There are still further challenges for all models to deal with. For instance, counting for objects requires intensive fine-grained recognition of the visual content demonstrated in the last row.

## 6 CONCLUSION

In this study, we presented Voila-A, a cutting-edge approach that aligns Vision-Language Models with user gaze attention, thereby improving efficacy in practical applications. Voila-A can be implemented in HMD AR/VR devices as an egoview copilot, benefiting a wide range of users, including visually impaired individuals who rely on their gaze to communicate their intent. This method surpasses the capabilities of similar mobile apps that necessitate users to lift their phones for scene capture. Despite its promising potential, there are limitations and future directions to consider. These include enhancing inference efficiency for real-time responses, integrating voice modalities for seamless interaction, and supporting higher resolutions for OCR and screen or UI understanding. We successfully utilized trace data to create the VOILA-COCO dataset, showcasing Voila-A's superior performance two benchmarks. Our research lays the foundation for more engaging human-AI interactions and encourages further exploration of Voila-A's integration with various modalities and tasks in the realm of multimodal AI systems.

## 7 REPRODUCIBILITY STATEMENT

In the interest of promoting transparency and facilitating further research within the community, we are committed to providing comprehensive resources alongside the publication of our work. To this end, we will release the VOILA-COCO and VOILA-GAZE datasets, as well as the annotation pipeline, training code, and model weights. By making these materials publicly available, we aim to ensure that our methods can be easily replicated and built upon by other researchers. Our commitment to reproducibility aligns with the broader scientific goal of fostering collaboration and the development of novel ideas and techniques in the field.

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

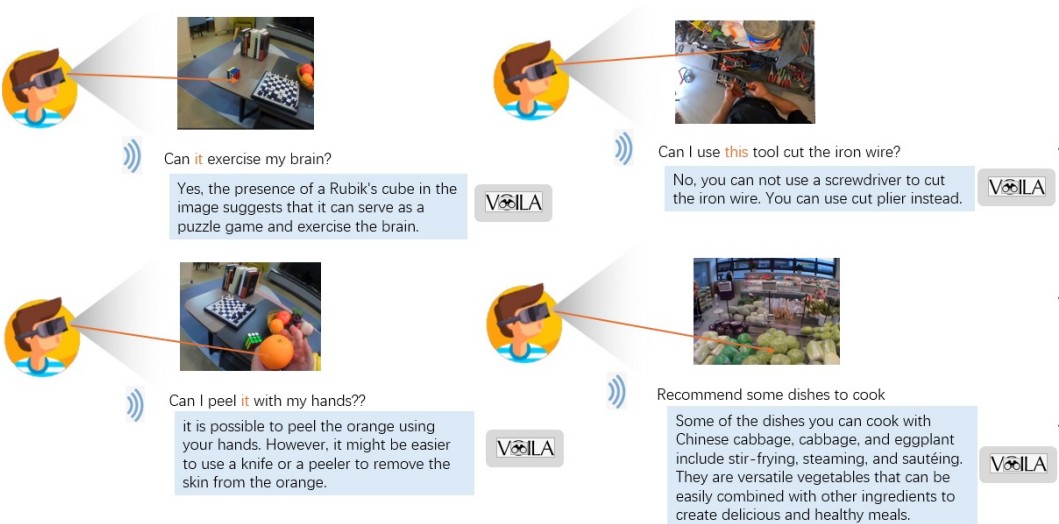

Figure 7: Different Scenarios in the Future

## A    LIMITATION AND DISCUSSION

### A.1    HALLUCINATION

During inference, our method occasionally exhibits hallucinations related to image content. This issue may stem from the limited number of training samples and the imperfect integration of visual hidden distributions into the language decoding process. Recognizing the potential for further scaling of our method, we consider addressing these hallucinations as a future research direction.

### A.2    COMPARISON TO GPT-4-V

With the release of GPT-4-V, which features vision capabilities, during the submission of this paper, we find it pertinent to include a brief discussion on the relationship between our work and GPT-4-V. Although GPT-4-V demonstrates remarkable visual capabilities in its demos, it surpasses our method in terms of visual understanding abilities. Nonetheless, our work remains valuable as it presents an effective approach to incorporating user sensory information for generating more relevant responses.

As reported, GPT-4-V occasionally struggles to accurately capture a user's intent when referencing specific elements within an image, prompting the design of an interface that allows users to directly draw sketches for highlighting purposes. Our method has the potential to enhance this user experience and can be extended to more dynamic scenarios such as virtual reality and augmented reality.

## B    GAZE DATA COLLECTION

In order to examine the gaze patterns of users, we conducted a preliminary study where participants engaged in one of three daily situations: grocery shopping, visiting a museum, or engaging in domestic activities. Participants were instructed to perform queries specific to each scenario, as if the system could provide the desired information. Our research involved 21 participants (8, 5, and 8 for each respective scenario), consisting of 13 males and 8 females, ranging in age from 19 to 30 years old (with a standard deviation of 3.18). The study took each participant between 90 and 150 minutes to finish, and they were compensated at a rate of 15 USD per hour for their involvement. As a result, we get 548 minutes of gaze recording. The Pupil Labs Invisible Tonsen et al. (2020) is a gaze-tracking smart glasses system that has been widely used for research purposes. It is equipped with gaze sensors, an egocentric camera, a microphone, and an inertial measurement unit (which was not used in this work). Participants were asked to wear the Pupil Labs Invisible glasses without

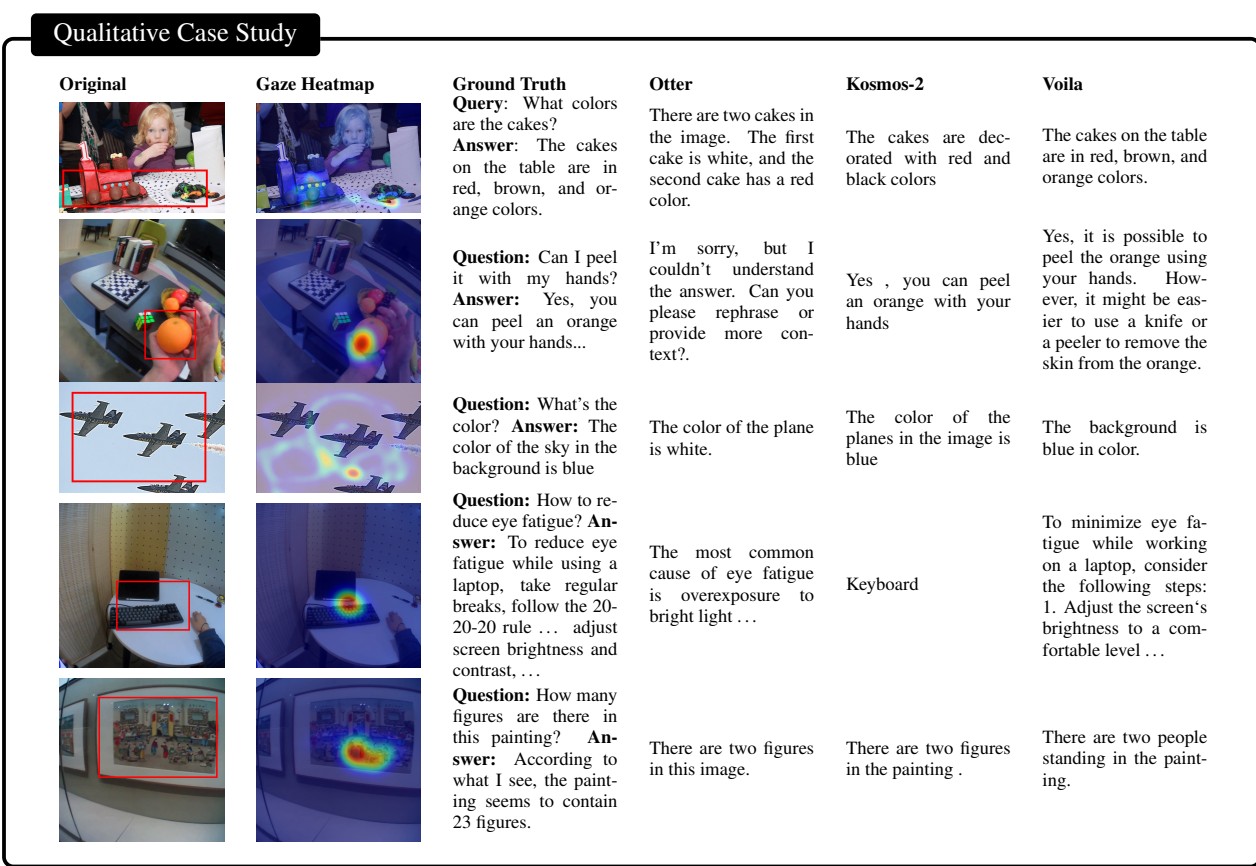

Figure 8: Qualitative Case Study: Top: We show successful predictions of all models. Middle: We demonstrate the problems of baseline models compared with VOILA including coreference queries, gaze grounding methods, etc. Bottom: We display hard challenges for all models.

| Supermarket shopping Task | Domestic living Task |
|---|---|
| Comparison | Appliance Malfunction |
| Completing Recipe | Activity & Health |
| Recommend | Snack & Fruits |
| Knowledge | Dressing Advice |
| Decision Making | Entertainment |
| Strengthen Decision | Small Talk |

Table 5: Guiding for User in VOILA-GAZE Collection, note this guide aims to facilitate and inspire users to generate questions related to data collection, rather than imposing strict limitations on the scope of their inquiries.

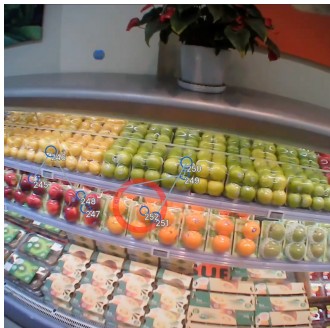
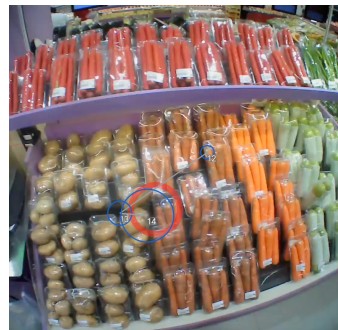
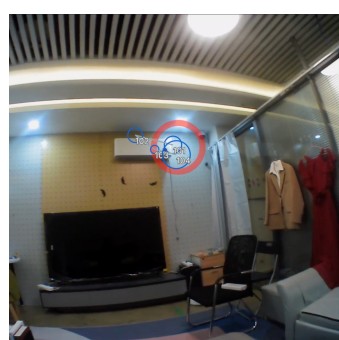

**Question: Please recommend some fruits for losing weight.**
**Answer**: To support weight loss, consider incorporating fruits like apples, oranges, and lemons into your diet. Apples are high in fiber and low in calories, which can help you feel full longer. Oranges provide a good source of vitamin C and fiber, while lemons can aid digestion and boost metabolism. Remember to consume these fruits in moderation and as part of a balanced diet for optimal results.

**Question: Which one is better for making dumplings?**
**Answer**: Potato is a better option for making dumplings, as it provides a smoother and creamier texture when cooked, making it ideal for dumpling fillings. Carrots can be used as an additional ingredient to add color and flavor, but using potatoes as the main ingredient will result in a more satisfying and traditional dumpling experience.

**Question: why is it not working?**
**Answer**: It's possible that your air conditioner is not working due to a variety of reasons, such as a dirty filter, low refrigerant levels, a malfunctioning thermostat, or electrical issues. To determine the exact cause, it's best to consult your user manual or contact a professional HVAC technician for a proper diagnosis and repair.

Figure 9: Data sample from VOILA-GAZE

any headwear that could obstruct the sensors on the glasses. Since the Invisible glasses require a connection to a mobile phone for operation, we instructed participants to keep the phone in their pockets to minimize potential distractions. Data was continuously recorded as participants engaged in their chosen scenario.

## C  DETAILS FOR AUTOMATIC DATA ANNOTATION

Figure 10 shows the system prompt and in-context examples for Automatic Data Annotation

## D  IMPLEMENTATION DETAILS

In this section, we describe the implementation details of our model. The model architecture is derived from Otter, which combines a text model and a vision model. The text model is an instance of MPTForCausalLM 7B and the vision model is based on the CLIP ViT-Large model with a patch size of 14.

- **Text Model Configuration:** The text model has 32 layers, each with 32 attention heads. The model has a hidden size and $d_{model}$ of 4096. The attention mechanism uses multi-head attention with torch implementation and alibi attention. The model uses learned position embeddings and low-precision layer normalization. The model is trained with a maximum sequence length of 2048 and a vocabulary size of 50432. The tokenizer used is EleutherAI/gpt-neox-20b. The model's torch data type is set to bfloat16.

---

**Prompt for Automatic Referring Annotation**

**System Prompt:** As an AI visual assistant, your task is to analyze a single view as if you are directly observing it. You will be provided with background and referable sentences describing the view, but your responses should be based on your observations rather than the descriptions. First, annotate the referable sentence using the <Q#></Q#> tag, where # is the tag number, to emphasize the section you want to discuss. Note: DO NOT annotate background sentences. Then, engage in a conversation with a person inquiring about the view, focusing on the tagged elements while considering the background information. Use the <Q#> tag for questions related to specific tags in the sentence, and the <Q> tag for other questions. The conversation should resemble the tone of a visual AI assistant examining the view and answering questions. Include a variety of questions and their corresponding answers. Your questions should cover various aspects of the view's visual content, such as object types, object count, actions, locations, and relative positions between objects. Ensure your questions have definite answers: (1) the view clearly displays the content in question, allowing for a confident response; or (2) it can be confidently determined from the view that the content is not present. Avoid questions that cannot be answered with certainty. Try your best to incorporate complex questions relevant to the view's content, including background information about the objects or discussions about events occurring within the view. Refrain from asking about uncertain details. When using <Q#>, follow the original question with an indirect question, as if the questioner is looking at the area of interest; use pronouns to refer to the relevant part, and omit some context in the original question if it can be inferred from the user's eye gaze fixation. When answering complex questions, provide detailed responses, including examples or reasoning steps, to create a convincing and well-organized explanation. Use multiple paragraphs if necessary. Make the answer comprehensive and informative. Throughout the conversation, maintain the impression that you are directly observing the view rather than reading descriptions. Do not mention any image or picture or text in the question and answer, as if you are equipped in a HMD and sharing the same view with the user.

- - - - - - - - - - - - - - - - - - - - - - - - - - - - - - - - - - - - - - - - - - - - - - - - - - - - - - - - - - - - - - - - - - - - - - - -

**In context Example:**

**Role: user**

Background: There is a movie theater that displays the show times above the doors.A hotel for dogs in the snow in winter.

Referable:A red fire hydrant is deep in the snow.The fire hydrant is in the snow near a recently plowed sidewalk.This city has had a very hard winter with snow.

**Role: assistant**

Refer:<Q1>A red fire hydrant</Q1> is <Q2>deep in the snow</Q2>.<Q3>The fire hydrant is in the snow near a recently plowed sidewalk.</Q3><Q4>This city has had a very hard winter with snow.</Q4>
===
Question:
<Q1>What is the red thing?
===
Indirect Question:
What is it?
===
Answer:
It is a red fire hydrant.
===
Question:
<Q2>Is the fire hydrant fully covered by snow or partially visible?
===
Indirect Question:
Is it fully covered by snow or partially visible?
===
Answer:
The fire hydrant is partially visible, with a significant portion of it covered by the deep snow.
===
Question:
<Q3>What is the condition of the sidewalk near the fire hydrant?
===
Indirect Question:
How's the condition of the sidewalk there?
===
Answer:
The sidewalk near the fire hydrant has been recently plowed, so it is relatively clear of snow compared to the surrounding area.
===
Question:
<Q4>What does this scene suggests about the winter season in the city?
===
Indirect Question:
How's the winter here?
===
Answer:
It suggests that the winter season in the city has been quite harsh and challenging. The significant snow accumulation seen in the view indicates that the city has experienced heavy snowfall, which can lead to various difficulties and disruptions for its residents.
For instance, the deep snow covering the fire hydrant could potentially hinder its accessibility in case of emergencies, which poses safety concerns. The recently plowed sidewalk implies that the city's maintenance crews have been actively working to keep the walkways clear and safe for pedestrians, but the sheer amount of snowfall might make it difficult to keep up with the continuous snow removal efforts.
Furthermore, such extreme winter conditions can impact transportation, leading to traffic issues, delays in public transit, and increased risks of accidents due to slippery roads. It can also cause problems for businesses and schools, as people might face challenges commuting to work or attending classes.
In conclusion, the red fire hydrant deep in the snow and the recently plowed sidewalk suggest that the city has faced a particularly severe winter season, with substantial snowfall that has likely caused various challenges and disruptions for its residents and infrastructure."

Figure 10: Prompt for Automatic Referring Annotation

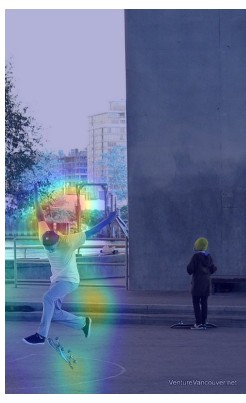 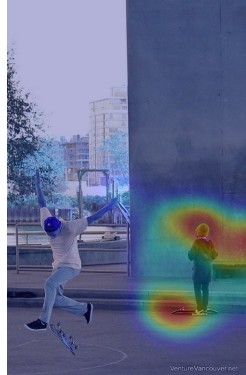 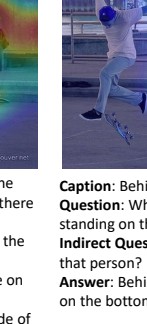

**Caption**: Bottom left side of the image a man is doing skateboarding and jumping.
**Question**: What is the skateboarder in the bottom left doing?
**Indirect Question**: What is he doing?
**Answer**: The skateboarder in the bottom left is performing a trick, jumping into the air with his skateboard.

**Caption**: Behind him there is a fencing. Behind the fencing there is water.
**Question**: What is behind the jumping skateboarder?
**Indirect Question**: What's behind him?
**Answer**: Behind the jumping skateboarder, there is a fencing , and behind the fencing, there is water.

**Caption**: Bottom right side od the image a person is standing and there is a skateboard.
**Question**: What can you see on the bottom right side of the view?
**Indirect Question**: What's there on the bottom right side?
**Answer**: On the bottom right side of the view, there is a person standing, and a skateboard is present near them.

**Caption**: Behind him there is a wall.
**Question**: What is behind the person standing on the bottom right?
**Indirect Question**: What's behind that person?
**Answer**: Behind the person standing on the bottom right, there is a wall.

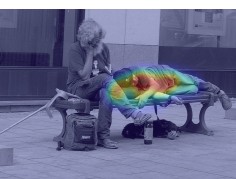 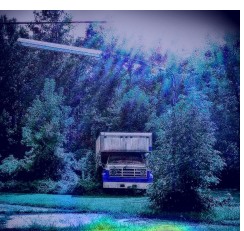 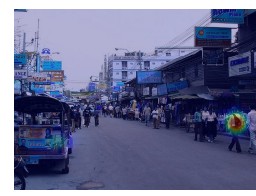 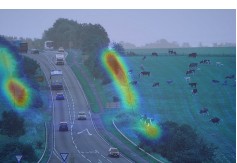

**Caption**: On the right, there is a man who is lying on his bag.
**Question**: What is the man doing on the bench?
**Indirect Question**: What is he doing?
**Answer**: The man is lying on his bag, possibly taking a nap or resting.

**Caption**: At the bottom of the image I can see the grass in green color.
**Question**: What color is the grass?
**Indirect Question**: What color is it?
**Answer**: The grass is green in color.

**Caption**: On the right side the woman wearing yellow colour shirt is walking wearing a black colour bag along with her.
**Question**: Can you describe the woman walking on the right side?
**Indirect Question**: What does she look like?
**Answer**: The woman walking on the right side is wearing a yellow shirt and carrying a black bag with her. She seems to be going about her day, possibly shopping or commuting.

**Caption**: There is a fence beside the ground.
**Question**: Is there any barrier between the road and the cattle's grazing area?
**Indirect Question**: Is there any separation between them?
**Answer**: Yes, there is a fence beside the ground, providing a barrier between the road and the cattle's grazing area.

Figure 11: Annotated Example of VOILA-COCO

---

**GPT Ranking**

**Overall System Prompt:** Given a question along with the ground truth description and answer of an image, evaluate the two provided candidate answers. Determine which answer is factually accurate, logical, and helpful to the user. if you think anwser 1 is better, respond with -1, if answer 2 is better respond with 1, if you think the result is tie, output 0. Only respond with either '-1' or '0' or '1' to indicate your choice.

- - - - - - - - - - - - - - - - - - - - - - - - - - - - - - - - - - - - - - - - - - - - - - - -

**Helpfulness System Prompt:** Given a question along with the ground truth description and answer of an image, evaluate the two provided candidate answers. Determine which answer is actually solves the user problem and more helpful to the user. if you think anwser 1 is better, respond with -1, if answer 2 is better respond with 1, if you think the result is tie, output 0. Only respond with either '-1' or '0' or '1' to indicate your choice.

- - - - - - - - - - - - - - - - - - - - - - - - - - - - - - - - - - - - - - - - - - - - - - - -

**Fact Grounding System Prompt:** Given a question along with the ground truth description and answer of an image, evaluate the two provided candidate answers. Determine which answer is factually grounded to the Fact provided. if you think anwser 1 is better, respond with -1, if answer 2 is better respond with 1, if you think the result is tie, output 0. Only respond with either '-1' or '0' or '1' to indicate your choice.

- - - - - - - - - - - - - - - - - - - - - - - - - - - - - - - - - - - - - - - - - - - - - - - -

```
 1: for each key in keys do
 2:     answer1 ← model1[key]["response"]
 3:     answer2 ← model2[key]["response"]
 4:     fact, gt_answer, question ← dataset[key]
 5:     prompt1 ← CreatePrompt(question, fact, gt_answer, answer1, answer2)
 6:     response_1 ← CallGPT(prompt1)
 7:     prompt2 ← CreatePrompt(question, fact, gt_answer, answer2, answer1)
 8:     response_2 ← CallGPT(prompt2)
 9:     score ← ComputeScore(response_1, -1 * response_2)
10:     Append scores with score
11: end for
```

Figure 12: GPT-RANKING Procedure

- **Vision Model Configuration:** The vision model has 24 hidden layers and a hidden size of 1024. It uses 16 attention heads and an intermediate size of 4096. The activation function is quick_gelu. The model uses an image size of 224 and 3 input channels. The patch size is 14, and the projection dimension is 512. The layer normalization epsilon is set to $1 \times 10^{-5}$.

- **Voila Configuration:** The architecture uses cross attention every 4 layers and only attends to previous layers. The model's torch data type is set to float32. Media placement augmentation is enabled during training.

For initialization, we use the Kaiming normal method with fan-in mode, ReLU nonlinearity, and a standard deviation of 0.02.

For optimization, we employ the AdamW optimizerKingma & Ba (2017) with a starting learning rate of 1e-5 and a batch size of 4. We train Voila for three epochs, scheduling the learning rate using a cosine annealing scheduler. To prevent exploding gradients, we apply gradient clipping with a threshold of 1.0.

## E  GPT-RANK

Figure 12 presents our prompt and evaluation procedure for GPT-RANKING

## F  ABLATIONS ON HOW TO INCORPORATE GAZE

Figure 13 shows different approaches exist for incorporating gaze data.

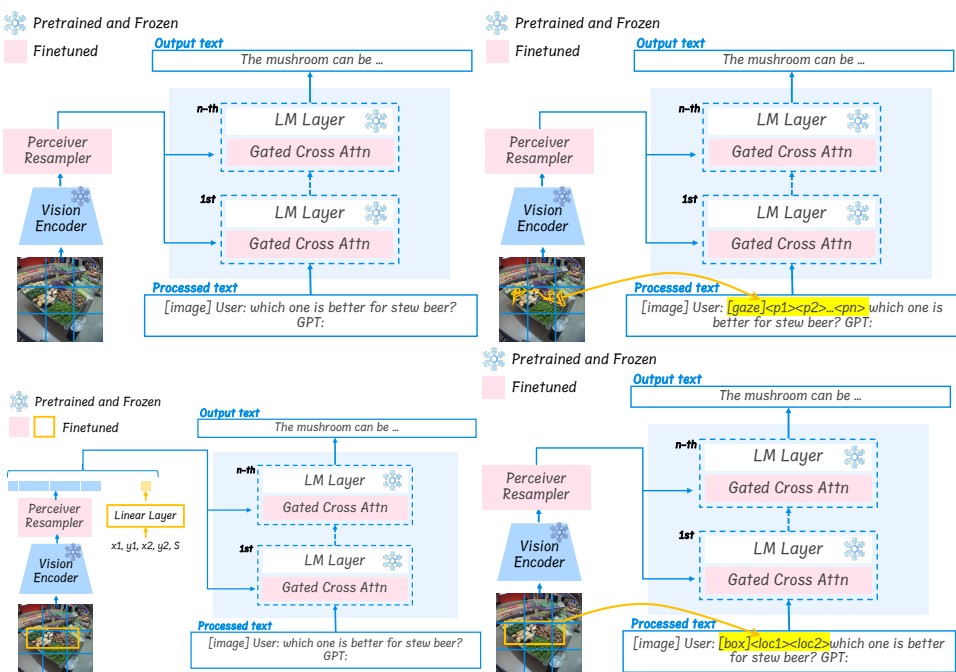

Figure 13: Different model design on how to incorporate gaze.Top-left: Otter-base, Top-right: Gaze as discrete position tokens. Bottom-left: Gaze *bounding box* as image patch Bottom-right:Gaze *bounding box* as discrete position tokens

## G  RELATED WORK

### G.1  MULTIMODAL LARGE LANGUAGE MODEL

Recent research works for multimodal vision and language tasks rely on multimodal large language models(MLLM) and demonstrated superior performance. One line is to learn **alignment** between the vision input to text token for LLM. LLaVA Liu et al. (2023) directly feed visual features to the LLM using only a learnable fully connected layer. BLIP-2 Li et al. (2023b) proposed Q-Former to extract a fixed number of query features from visual features, which are aligned with the text in frozen LLM. Another direction is to design **learnable interaction layers** to attend the vision input to the frozen LLM layers. FlamingoAlayrac et al. (2022) adopt perceiver resampler module to convert visual features into visual tokens and interleave the tokens in plain text at the locations of vision inputs. Besides, Flamingo performed learnable cross attention to attend to the visual tokens of the image that appeared just before it in the interleaved sequence, rather than to all previous images and built a large-scale interleaved image-text dataset. On top of these design, recent works mainly focus on improving **instruction-following ability**. LLaMA-AdaptersZhang et al. (2023a) aims to adapt LLaMATouvron et al. (2023) into an instruction following model with an additional adapters module and multi-modal prompts. Mini-GPT4 Zhu et al. (2023), mPLUG-OWL Ye et al. (2023), and InstructBLIP Dai et al. (2023) adopt the Q-Former on various language models for instruction following capability. Besides, MultiModal-GPTGong et al. (2023) fine-tuned OpenFlamingoAwadalla et al. (2023) using Low-rank Adapter (LoRA)Hu et al. (2021) and OtterLi et al. (2023a) introduced MultI-Modal In-Context Instruction Tuning (MIMIC-IT) dataset following three heuristics, both of which demonstrate improved instruction-following ability, Simultaneously.

**Grounded MLLM** Inspired by the success of MLLM, recent works focus on investigating the fine-grain grounding capability between region-text pairs instead of image-text pairs, and further conduct dense regional prediction tasks. One research line is to learn **regional alignment** between the image regions with the corresponding text tokens. RegionCLIPZhong et al. (2022) extends CLIP with pseudo image regions and textual concepts pairs. GrillJin et al. (2023) proposes to replace the referring words with the corresponding visual patches to align text and image Regions. RegionBLIPZhou et al. (2023) takes position-assisted regional objects as soft prompts for LLM on image-region-text

data. Another research focus is to unleash the **grounding ability** in a multimodal large language model. VL-T5Cho et al. (2021) converts the visual grounding task into regional *ROI box* feature conditioned text generation to predict the box id. OFAWang et al. (2022), PEVLYao et al. (2022) and KOSMOS-2 Peng et al. (2023) reformulate continuous corner coordinates of object to *discrete position tokens*. ShikraChen et al. (2023) handles *spatial coordinate* inputs and outputs in natural language without introducing extra vocabulary or position encoders. The worksChen et al. (2023); Peng et al. (2023); Zhang et al. (2023b) also perform Instruction tuning and convert the position of regional objects into language descriptions. Although gaze is flexible and interactive, it is easy for humans to understand the gaze's semantic representation but hard for AI agents.

## G.2 REGION REPRESENTATION FOR LARGE LANGUAGE MODELS

The visual region can be represented as **bounding boxes**Zhu et al. (2016); Liu et al. (2017), **points** Mani et al. (2020), **traces**Pont-Tuset et al. (2020); Yan et al. (2021). Existing approaches usually leverage Fast-RCNN to detect bounding boxes which limits the pre-defined or recognized objects in the bounding box and hard to scale out. Points are flexible but are too fine-grain and require large amount of points to represent large regions precisely. Trace is a more natural way to input by using the mouse trace coordinates and most similar to human gaze. In AR and VR scenario, although trace is applicable with gesture, we propose to use gaze as a more convenient and interactive way. The two worksChen et al. (2023); Mani et al. (2020) takes bounding boxes or points as region input for visual question answering and are the most similar work. Different from them, we take **gaze** as regional inputs.

**Region Inputs** In order to input the regional information to the model, several methods(Zhang et al., 2023b; Bracha et al., 2023) directly **concatenate cropped image patches** with the original text/image as model input. Another methods (Lin et al., 2020; 2022) use 0/1 **mask or Gaussian map** input with the original image to emphasize the area of user interest. Additionally, other methods Kirillov et al. (2023); Voigtlaender et al. (2023) first encode points, boxes or trace to **positional encodings** then add them to intermediate features or learned queries. Specifically for gaze, Qian et al. (2023) propose a gaze-directed visual grounding and *fuse* the gaze feature through a multi-modal fusion module. EG-ViT(Ma et al., 2023) propose a eye-gaze-guided vision transformer which takes the *masked image patches* within the gaze interest.

## G.3 GAZE AND CURSOR AS A PROXY FOR ATTENTION

Cursor-based techniques, including approaches like SALICON(Jiang et al., 2015) and Bubble-View(Kim et al., 2017), have emerged as affordable, nonintrusive, and scalable alternatives to traditional eye-tracking methods for collecting human attentional data. Empirical evidence from prior work has established strong connections between cursor-like signals and gaze positions. Studies focusing on web browsing and search tasks have found a high correlation between cursor and gaze locations, with better alignment along the vertical dimension(Huang et al., 2011; 2012; Guo & Agichtein, 2010; Chen et al., 2001). These findings support the motivation to use cursor-based techniques as a proxy for attention.

Despite their success, existing cursor-based studies have limitations, such as the need for complex post-processing of mouse movement data and evaluations limited to simple aggregate comparisons with eye-tracking data(Kim et al., 2017). Furthermore, while prior work serves as a solid foundation from a data-centric perspective, it lacks demonstration of whether modern applications aiming to assist users using gaze, such as vision language models (VLMs), can be trained from cursor data and later adapt to gaze signals, especially when transitioning from 2D planar images on screens to ego view scenes in head-mounted display (HMD) scenarios. Our work aims to directly tackle this problem, as we believe it is the optimal time to close the entire visionary loop of understanding and utilizing the gaze modality to ultimately achieve smart, in situ personal assistants.

## G.4 SALIENCY MODELS ON MODELING GAZE ATTENTION

Yarbus (1967) proposed that tasks could be decoded from fixation patterns, receiving mixed support in subsequent research. Early computational models of visual attention focused on bottom-up approaches, representing pre-attentive selection processes from visual input(Koch & Ullman, 1985).

Later, the saliency map concept emerged(Niebur & Koch, 1995). Initially, models were trained on fixation data from eye-tracking experiments(Kienzle et al., 2006; Judd et al., 2009), but collecting large datasets proved difficult. The SALICON dataset(Jiang et al., 2015) addressed this challenge by using mouse movements to simulate natural viewing behavior, leading to state-of-the-art performance in saliency models (Jiang et al., 2015; Pan et al., 2016; Tavakoli et al., 2017). As deep learning advanced, saliency modeling improvedKümmerer et al. (2016), enabling more complex gaze pattern modeling in vision-language tasks(Sugano & Bulling, 2016; Das et al., 2016; Vasudevan et al., 2018).

**Saliency models aim to approximate the human visual system by predicting eye fixations on images** (Kim et al., 2017). Unlike traditional saliency models, **our approach takes ground truth gaze data, image, and natural language inputs to generate contextually relevant responses**, presenting a novel challenge. Recent work, such as Sugano & Bulling (2016), leverages gaze signals to enhance captioning tasks but does not accommodate dynamic user queries beyond captioning. Additionally, their LSTM-based method falls short compared to contemporary large transformer baselines. Sood et al. (2021) introduces gaze data to the visual question-answering (VQA) task, but their analysis remains limited to comparing human and neural attentive strategies learned by VQA models. With the development of large vision-language models, we believe our work provides a valuable contribution to modern applications by effectively tackling the new challenge.

