# OpenReview forum: "Voila-A: Aligning Vision-Language Models with User's Gaze Attention"
_ICLR.cc/2024/Conference — Submitted to ICLR 2024_

### Official Review · Reviewer_gfYi · 2023-10-23

**Soundness:** 3 good
**Presentation:** 2 fair
**Contribution:** 2 fair
**Rating:** 5
**Confidence:** 3

**Summary:**

This paper introduces the concept of incorporating gaze information (referred to as human attention) into VLM models to enhance their performance and potentially improve their interpretability. To support this, the authors collected hundreds of minutes of gaze data and developed an automated data annotation pipeline using GPT-4 to generate the VOIA-COCO dataset.

The paper makes a valuable contribution to the relevant academic community by demonstrating the significance of gaze information for VLM tasks, such as visual question answering. The presented results show that the proposed method is qualitatively and quantitatively superior to baseline models like Otter.

**Strengths:**

•	The authors introduce the novel concept of utilizing gaze information in the development of VLMs.
•	Unlike baseline and several other studies, the experimental analysis is not limited to qualitative results but also demonstrates quantitative results.

**Weaknesses:**

1.	The paper's presentation is lacking. Many important sections have been relegated to the appendix, especially the technical details. For example, Section 4.1 is challenging to understand due to the limited text. 2)The model heavily depends on the baseline model Otter. The method of injecting gaze information is quite straightforward. The way in which the authors handle catastrophic forgetting can be observed in the literature, thus not introducing technical novelty.
2.	The experimental analysis appears somewhat unfair because the proposed method uses additional modalities to achieve the same results. Therefore, its better performance is not surprising, particularly in cases where the query does not clearly define the object's name, and several other objects are present in the scene, with the gaze heatmap aligning with the queried object. It is also worth to mention that both baseline methods are still only in ArXiv.
3.	There is uncertainty about the cases in which gaze information was found to be less relevant.
4.	The caption for Figure 3 lacks informativeness.
5.	There exist a few typos to be fixed, e.g., Fiture
6.	Hallucination issue can be better presented qualitatively and better discussed.
7.	Gaze data collection procedure is also very scarse and it is not possible to understand if the annotators have a reliable consensus to use the collected data in model evaluation and comparisons.
8.	It is doubtful whether 100 gaze samples are sufficient for conducting a comprehensive comparative study. I have reservations about the potential bias in the collected dataset.

**Questions:**

•	Section 4.1 and gaze data annotation should be described in detail. It is not possible to validate the procedures perform in these context.
•	Weakness Q4
•	Pls. comment on Weakness (2) for the technical novelty of the method.
•	Pls. comment on Weakness (3).
•	How the authors evaluate the interpretability? Several places in the paper interpretability was mentioned, however its evaluation is unclear given that such a keyword is being used in several different content of AI.
•	Weakness Q9

---

> ### Author Response · Authors · 2023-11-19
> **Response to Reviewer gfYi (1/2)**
>
> Thank you for your insightful feedback on our work. Your input is invaluable in helping us identify improvements and refine our research. We're committed to addressing any shortcomings and appreciate your constructive suggestions.
>
> **Presentation and Technical Details:**
>
> We recognize the need to enhance our presentation and include crucial information in the main paper by providing detailed explanations where needed. To address your concerns, we have made significant improvements to our manuscript, including extensive reorganization, clarification, and rewriting as we replied in **General Response**.
>
> **Baseline Selection and Technical Novelty**
>
> We would like to begin by clarifying the rationale behind our choice of Kosmos-2 and Otter as the two baselines for comparison. Our method aims to demonstrate improvements in two key areas: state-of-the-art multimodal conversation capabilities and grounding ability with additional modality input.
>
> First, Otter is based on the Flamingo architecture, a well-recognized milestone in vision-language research. Otter has been fine-tuned on egocentric data (e.g., Ego4D) and instructional data (e.g., COCO), making it more suitable for the scenarios we are targeting compared to other similar VLMs. Second, Kosmos-2, at the time of submission, is a widely known and comprehensively evaluated model with superior grounding capabilities. Since our model has not been pre-trained on massive grounding data like Kosmos-2, comparing our grounding capability to Kosmos-2 presents a significant challenge, requiring us to explore efficient ways to guide VLMs in learning grounding.
>
> Regarding the reliability of our chosen baselines, both Kosmos-2 and Otter are cited extensively (72 and 110 citations, respectively), open-sourced, and actively maintained. We believe that the research community widely recognizes and reviews these models.
>
> In terms of fairness in comparison to the baselines, for grounding, we contend that our method performs on par with, if not surpassing, Kosmos-2, which also has an additional modality indicating user focus and pre-trained on a large grounding dataset. For Otter, our method not only demonstrates significant grounding improvements but also exhibits better helpfulness. The quality of our coreference question responses is comparable to the direct question responses of Otter's base model, which we believe is a fair comparison.
>
> As for technical novelty and contribution, we argue that our model design is far from trivial. Numerous approaches exist for incorporating gaze data, and our final choice was made after conducting a series of experiments detailed in the ablation study. To further elaborate on these aspects, we have included a detailed illustration of different gaze integration methods in Figure 13. Additionally, our work extends beyond model design, as we strive to establish a method for a new problem. The data annotation pipeline and gaze benchmark construction have not been explored by other works, making it difficult to claim that our work lacks novelty.
>
> **Interpretability**
>
> We mentioned interpretability twice in our conclusion, as we believe integrating more human intention signals into the model can enhance mutual understanding between the model and the user. This means the model can better interpret the user's intent based on the additional modality, and the user can understand the model's specific response according to the pre-input gaze grounding signal. However, we acknowledge that using the term "interpretability" might lead to confusion with the traditional interpretation concept in machine learning. Therefore, we have decided to temporarily remove this term and reserve it for further evaluation and demonstration. We genuinely appreciate your valuable feedback on this matter.

---

> > ### Author Response · Authors · 2023-11-19
> > **Response to Reviewer gfYi (2/2)**
> >
> > **Gaze Data Collection Procedure and Sample Size**
> >
> > In the revised version, we have expanded upon the gaze data collection procedure and provided further information. We appreciate your concerns regarding potential bias in the dataset and the sufficiency of the sample size, and we are more than willing to engage in a discussion on these matters:
> >
> > 1. We acknowledge that bias in any test set is inevitable when the sample size is limited. However, the primary concern should be whether the potential bias significantly impacts the purpose the set is intended to serve.
> > 2. The main motivation for collecting VOILA-GAZE is to evaluate the adaptability of trace-trained VLMs to gaze-directed downstream use cases without a substantial decline in performance.
> > 3. While our dataset's primary limitation is the confined range of scenarios, which may not fully assess a large VLM's knowledge and capabilities, we have made an effort to include a diverse group of participants in terms of gender, age, major, and other factors to address gaze behavior.
> > 4. Based on both quantitative and qualitative experiments, we are confident that the improvements brought about by our pipeline design are both significant and reliable.
> > 5. We are actively working to expand and enhance this benchmark by incorporating a broader range of application scenarios, which will further strengthen the validity of our findings.
> >
> > We hope this clarification addresses your concerns, and we look forward to any further discussion on this topic.
> >
> > **When Gaze is Less Relevant**
> >
> > You raise an important point regarding the impact of gaze relevance on user experience in real-life scenarios. We would like to address this in two parts. First, when a user's query is clear enough, we observe that models tend to prioritize language instructions in most cases. This can be seen in our ablation studies on coreference and direct query ablations. This behavior may be attributed to the presence of noise in the training data, where gaze traces are not correctly aligned, causing the model to learn robustness in such cases.
> >
> > Second, when a user's query is unclear, the model may attempt to connect the irrelevant gaze with the query, potentially leading to inaccurate responses. We acknowledge that our current observations are limited to qualitative cases, and we plan to evaluate this aspect quantitatively as we refine and iterate on our method in the future.
> >
> > **We hope our responses have sufficiently addressed your questions and concerns. If you believe we have met your expectations, we would be grateful if you could consider increasing your score to further support our work. Thank you!**

---

### Official Review · Reviewer_2Muy · 2023-10-31

**Soundness:** 2 fair
**Presentation:** 2 fair
**Contribution:** 3 good
**Rating:** 6
**Confidence:** 3

**Summary:**

This paper presents "Voila-A," a novel approach aimed at aligning Vision-Language Models (VLMs) with user gaze attention. The authors highlight the challenges faced by existing VLMs in handling complex scenes and diverse human attention patterns. They propose utilizing gaze information collected from AR or VR devices as a proxy for human attention to guide VLMs.

The paper provides a thorough explanation of the methodology, including data collection, automatic data annotation using GPT-4, and the design of the Voila Perceiver modules. The authors conduct experiments, comparing Voila-A with baseline models (Otter and Kosmos-2) on both synthesized and real-world gaze datasets.

The results demonstrate the effectiveness of Voila-A, showcasing its balanced capability between helpfulness and fact grounding. The evaluation metrics, including GPT-4 Ranking and Reward Score, support the authors' claims. Additionally, ablation studies and qualitative analyses provide further insights into the model's performance and capabilities.

One notable contribution is the introduction of trace data as a proxy for gaze data, offering a cost-effective and scalable alternative for aligning VLMs with user gaze attention. The method of transforming trace data to mimic gaze data is well-described and substantiated with empirical evidence.

**Strengths:**

The proposed method for aligning gaze data with trace data proves to be both effective and straightforward. It introduces a fresh approach to integrating gaze information with Vision-Language Models (VLLM). The approach has been rigorously examined through studies, yielding results that substantiate its efficacy.

**Weaknesses:**

The data collection section (4.1) lacks detailed information on the methodology and content of the dataset. Providing specific examples and clearer explanations would enhance comprehension. Additionally, Figure 3 needs a caption and more in-depth explanations to convey its intended message. The figures also need higher resolution for better readability when printed.

Section 4.2 requires clearer explanations, particularly regarding the parameters X, L, and G. The concept of 'latent data' (L) needs better elucidation. A structured approach, starting with an explanation of the inputs and employed encoders, followed by a deep dive into the new approach, would enhance clarity. A comprehensive figure illustrating how gaze is integrated into visual language models would be beneficial. It's unclear how the 'Voila Perceiver Resampler' module is integrated with the VLLM.

In Section 4.3, the meaning and specifics of 'gaze linear layer' and 'gaze key linear of attention' need clarification. It's not clear which layers these terms refer to or if there's a specific formula involved.

Merging Section 5.1.2 with Section 4.1 would improve the overall clarity of the paper.

The summary of the main results in Figure 5 is not easily understandable. Using a table with percentages might provide clearer insights.

The claim that Voila exhibits a 'superior ability to handle multi-turn real conversations' in the last sentence of Section 5.3 needs stronger support or clarification in the results section.

**Questions:**

Could you provide a clearer depiction of how the ViolapercieverBlock and Resampler are integrated within the larger VLLM framework? A simplified architectural overview would be immensely helpful in understanding the bigger picture.

It would be beneficial to have more details on what exactly is included in the automatic data annotation process and how it is carried out. Providing specific examples in the main paper would greatly enhance comprehension.

For Figure 4 and 5, additional guidance on how to interpret the results would be appreciated. Specifically, clarification on what constitutes the 'Overall score' and a detailed explanation of how the 'Helpfulness' and 'Grounding score' are calculated would be invaluable.

I would also onsider providing a brief discussion of potential applications and future directions in the conclusion section.
Clarify any specific limitations or potential challenges associated with the proposed approach.

---

> ### Author Response · Authors · 2023-11-19
> **Response to Reviewer 2Muy**
>
> We are sincerely grateful for your positive assessment of our work and appreciate the constructive and valuable suggestions you've provided to help us improve it further.
>
> **Automatic data annotation process in Section 4.1**
>
> We acknowledge that our original presentation of this process was not sufficiently clear, making it challenging for readers to follow. In the revised version, and in accordance with your recommendations, we have provided detailed formulations and explanations of the annotation process alongside Figure 3. We have also included final annotated data examples in Figure 11 to enhance comprehension. Furthermore, to facilitate better viewing, we have replaced the original PNG figure with a PDF format, allowing readers to zoom in as needed.
>
> **Voila Perceiver and Resampler Explanation**:
>
> In the revised version, we have provided a clearer architectural overview to better illustrate how the Voila Perceiver and Resampler components are integrated within the VLM framework. Additionally, we have marked the different fine-tuning stages on the structure to clarify which components are tuned in each phase. We acknowledge that the original terms `gaze linear` and `gaze attention` were not clear, and we have revised the text to specify that they refer to the components tuned during the first stage, without introducing any new formulations. Furthermore, we have rewritten the introductory part of the method section to provide more context on our base structure and input, ensuring a more comprehensive understanding for our readers.
>
>
>
> **Interpretation of Figures 4 and 5**:
>
>  In the revised manuscript, we have provided additional insights on interpreting the results, along with explanations for the 'Overall score,' 'Helpfulness,' and 'Grounding score,' which can be found in Figure 12 in the appendix. The primary distinction between these scores is as follows:
>
> 1. For Helpfulness, the ground truth answer is provided as a reference, allowing GPT-4 to determine which model's output effectively addresses the user's question.
> 2. For Grounding, the original captions from COCO and Localized Narratives are given, enabling GPT-4 to decide which one aligns better with the actual content of the image.
> 3. The Overall score involves both ground truth captions and answers as input, allowing GPT-4 to evaluate which of the two models' answers is superior in general.
>
> **Conclusion section**:
>
> In the revised conclusion section, we have incorporated a discussion on potential applications, future directions, and limitations of our method. Specifically, our approach can be employed in HMD AR/VR devices to serve as an egoview copilot for everyone, including visually impaired individuals who may not have clear vision but can still use their gaze to convey their intent. Our method offers greater applicability than similar mobile apps that require users to raise their phones to capture a scene for assistance.
>
> The limitations and future directions include:
>
> 1. Improving inference efficiency to enable instantaneous responses.
> 2. Incorporating voice modalities to facilitate seamless interaction.
> 3. Supporting higher resolutions to accommodate scenarios requiring OCR and screen or UI understanding.
>
> Additionally, we have addressed the hallucination problem and discrepancies with GPT-4V in the appendix. By discussing these aspects, we aim to provide a comprehensive understanding of our method's potential and areas for future improvement.
>
> **Other Issues**
>
> 1. We have merged Section 5.1.2 with Section 4.1 to streamline the presentation of the related content and improve the overall flow of the manuscript.
>
> 2. Regarding the phrase "superior ability to handle multi-turn real conversations," we have updated the statement to clarify its meaning and avoid any potential misunderstandings. The updated statement emphasizes that when appending more in-context conversations, our method's performance continues to improve compared to single-turn baselines. This change ensures that readers have a more accurate understanding of the method's capabilities.
>
> **We hope that our responses have adequately addressed your questions and concerns. If so, we would greatly appreciate it if you could consider increasing your score to further support our work. Thank you!**

---

### Official Review · Reviewer_428q · 2023-11-01

**Soundness:** 1 poor
**Presentation:** 2 fair
**Contribution:** 2 fair
**Rating:** 3
**Confidence:** 2

**Summary:**

This paper introduces a way to integrate gaze information into large VLMs. It uses mouse trace data as a proxy for gaze track with proper sampling. Such gaze information is then used in an attention mechanism to enhance visual feature perception. The authors report that the proposed approach outperforms baselines.

**Strengths:**

* Introduces a scalable way to leverage human attention cue in VLM models.

**Weaknesses:**

* Technical or scientific contribution is very incremental and limited.
* Writing can be improved; not always easy to follow and clear.
* Baseline methods considered are not comprehensive or fair. Mouse trace data as a proxy for gaze sounds reasonable but there are many off-the-shelf saliency model that are designed to mimic human gaze. Some of the existing saliency model can be used or at least need to be discussed and reviewed in the paper.

**Questions:**

Please address the comments above regarding baseline.

---

> ### Author Response · Authors · 2023-11-19
> **Response to Reviewer 428q**
>
> We would like to express our gratitude to Reviewer 428q for your astute and concise critique of our work. While we appreciate your valuable feedback, we believe that there may be some misunderstandings about certain aspects of our study. To address your concerns, we have provided detailed explanations below and welcome the opportunity for further discussion:
>
> **Baseline Methods:**
>
> We are grateful for your suggestion to discuss saliency models in our paper. In the revised version, we have included a dedicated discussion of saliency models in Section G.4 and clarified their relationship with our proposed method. However, we would like to emphasize that saliency models, which predict human gaze fixation from images, cannot be directly compared to or used in our approach as baselines. In our task, the ground truth gaze, image, and user query are used as inputs to generate a response, which is a different problem setting.
>
> The most relevant work in this context involves using gaze as input for generating captions, such as the work by Sugano and Bulling [1]. However, these methods predate the era of large-scale language models and are designed for single-purpose tasks like captioning. As they have not been pretrained on massive datasets, comparing their performance to our modern approach would be unfair, as their performance is undoubtedly lower than ours.
>
> Instead, we have conducted an ablation study to investigate different aspects of the model design space, such as input types, gaze integration methods in VLMs, tuning techniques, and data variations. We believe that these analyses provide a more meaningful baseline for comparison and offer valuable insights into the impact of various design choices on performance. And we have update a clear illustration of those baselines in Figure 13, it may be helpful for you to check.
>
> [1] Yusuke Sugano and Andreas Bulling. Seeing with humans: Gaze-assisted neural image captioning. ArXiv, abs/1608.05203, 2016.
>
> **Writing and Presentation:**
>
> In response to all reviewers' suggestions, we have extensively reorganized, clarified, and rewritten the manuscript, as well as expanded the material in the revised version as we replied in general response. We hope that these changes address your concerns effectively. We remain open to further suggestions and would appreciate any additional feedback on areas that may require improvement or clarification.
>
> **Technical Contribution**
>
> We respectfully disagree with your assessment of our work's technical valuation and contribution, and we would like to provide the following reasons for your reconsideration:
>
> 1. *Motivation:* As discussed in the revised version, although there is existing research on gaze-related user interfaces and gaze behavior modeling such as saliency models, there has been no investigation into integrating gaze data into modern models and applications. Our work pioneers this area by providing a scalable solution to address this challenge, paving the way for further advancements with high potential.
>
> 2. *Effort and Innovation:* We are the first to define a new gaze-enhanced assistance task that is crucial for the upcoming era of spatial computing. We have invested considerable effort in building a reliable automated data annotation pipeline and providing guidance on potential issues that may arise during this process. Additionally, we have developed a comprehensive solution for scalable training and evaluation of models, established new benchmarks using real-life user data, and offered various insights into model design and training through extensive ablation studies. The work presented is far from trivial and should be recognized as a significant contribution.
>
> 3. *Contributions to the Research Community:* We plan to release both datasets, the automated data annotation pipeline, pretrained model weights, and training code. Given the challenges in obtaining gaze collection equipment, invoking massive APIs for automated annotation, and fine-tuning and evaluating comprehensive ablations on models with billions of parameters, we believe the resources we are providing will be valuable and directly enhance future research in this area.
>
> **In light of the points mentioned above, we kindly request that Reviewer 482q reconsider the rating for this work. We also welcome any further questions or suggestions you may have.**

---

### Official Review · Reviewer_tDrZ · 2023-11-06

**Soundness:** 3 good
**Presentation:** 2 fair
**Contribution:** 3 good
**Rating:** 6
**Confidence:** 4

**Summary:**

In AR/VR scenarios, gaze is one of the most natural way to represent the regions interesting to users. This paper studied an interesting problem: suppose we are using vision-language model under AR/VR, how to incorporate human gaze attention into vision-language model and how much improvement can it bring? The paper proposed to use mouse trace to approximate gaze and use the collected gaze heatmap into attention mechanism in vision language models (Otter) while freezing the language encoder MPT-7B and vision encoder CLIP ViT-L/14. The models is evaluated on the collected Voila-COCO data set and a VOILA Gaze data which is more close to real life scenarios. The proposed method with extra gaze information outperforms baselines Otter and Kosmos-2. Ablation study also shows that gaze heatmap is better than alternatives ways to use gaze data like discrete gaze position, gaze bounding box as patch, etc.

**Strengths:**

--The idea of including gaze information to vision-language model is quite interesting, which might be one important aspect when people use the vision-language model in VR/AR scenarios. The idea of human using gaze/attention in compute vision models is not new, but the idea of using gaze/attention to improve vision-language model is relatively novel to the best of my knowledge.

--Some interesting experiment results are shown to demonstrate that the gaze/attention data are helpful for VQA tasks of vision-language models.

**Weaknesses:**

--Will the data set VOILA-COCO be released? I did not see this information in the paper.

--Using mouse trace to approximate human gaze/attention is a popular approach in attention related area, however, the authors does not mention existing works like BubbleView https://bubbleview.namwkim.org/ or Salicon http://salicon.net/

--The organization and presentation of the paper can be improved. It is not clear how the gaze data will be used in vision-language model until section 4.3. Instead, the authors can provide an illustrator figure about it at the beginning.

**Questions:**

See the weakness part. Especially, the dataset might be an important contribution of this paper. However, it is not clear whether the data set will be released or not.

**Details Of Ethics Concerns:**

Human gaze data are collected and used.

---

> ### Author Response · Authors · 2023-11-19
> **Response to Reviewer tDrZ**
>
> We appreciate your recognition of our work's ideas and experiments. We would like to address your concerns as follows:
>
> **VOILA-COCO dataset release**
> We thank the reviewers' interest in the VOILA-COCO and VOILA-GAZE datasets. In response to their inquiries, we would like to confirm our intention to **release both two datasets** upon the publication of our work. Our aim is to foster further research and collaboration within the community by providing these valuable resources. In addition to the datasets, we will also make our annotation pipeline, training code, and model weights publicly available, ensuring that our work is transparent and easily reproducible. To address this concern explicitly, we have included a reproducibility statement in the revised manuscript. We hope that these efforts will contribute positively to the field and facilitate the development of new ideas and methods.
>
> **Ethical Considerations**
>
> Our experiments have been reviewed and approved by the Institutional Review Board (IRB) at our institute. All participants have provided informed consent, agreeing that their data will be used as part of a publicly available dataset. To ensure privacy, all personal information has been properly anonymized. Additionally, we will conduct a thorough review of any potential privacy issues before releasing the dataset to the public. We are commit to upholding ethical standards in research and protecting the privacy of our study participants.
>
> **Existing works on mouse trace approximation**
>
> We sincerely appreciate your suggestion to consider the relevant background literature on mouse trace approximation. It is encouraging to discover that, despite differing motivations and experimental settings, these works provide robust support for some of our hypotheses. In response to your feedback, we have conducted a comprehensive survey and included an overarching discussion in Section 2, as well as detailed analyses from various perspectives in Appendices G.3 and G.4.
>
> In our revised manuscript, we have incorporated references to prominent works such as SALICON and BubbleView, and provided a comparative analysis to elucidate the connections and distinctions between our approach and these existing methods. We believe this addition strengthens our paper and offers valuable context for readers. Thank you once again for your valuable input.
>
> **Improved Organization and Presentation** We recognize that the paper's organization and presentation could benefit from enhancements. In the revised version, we have introduced a comprehensive Figure 4 to elucidate the utilization of gaze data in vision-language models. Furthermore, we have restructured the sections pertaining to data collection and result analysis to improve readability and facilitate a better understanding for our readers.
>
> **We hope that our responses effectively address your questions and concerns. If so, we kindly request that you consider raising your score to further support our work. Thank you!**

---

> > ### Comment · Reviewer_tDrZ · 2023-11-22
> > **Resolved most of my concerns**
> >
> > Increase my rating

---

### Author Response · Authors · 2023-11-19
**General Response**

Dear Reviewers:

Thank you for taking the time to review our submission and providing valuable feedback. We appreciate your comments and suggestions, which have helped us identify several areas for improvement. In this rebuttal, we address the concerns and questions raised by the reviewers and provide clarifications where needed. We have made significant changes to our manuscript and highlighted those changes in blue for your convenience:

1. We have reorganized Section 2, replacing many basic explanations with a comprehensive review of different types of gaze-related prior work. We also refer to additional detailed related work discussions in Appendix G.3 and G.4, focusing on the relationship between cursor-based methods and gaze attention, as well as saliency models, respectively.

2. We have consolidated all data collection information into Section 4, providing detailed and structured narrations alongside Figure 3: Automatic Data Annotation Pipeline. We have also enhanced the description of the VOILA-GAZE collection process in Section 4.2.

3. We have included data samples of VOILA-GAZE and VOILA-COCO in Figures 9 and 11, respectively.

4. We have added an overall model structure of the VOILA-MODEL, featuring a zoomed detail of the VOILA PERCEIVER to clarify the relationships between components and illustrate the fine-tuning steps using a color legend.

5. We have introduced the background of our model design in Section 4.3 and presented all potential structural ablations we have compared in Figure 13, demonstrating that our model design choice is not trivial.

6. We have updated the interpretation guide for our main results according to reviewer feedback.

7. We have revised our qualitative studies with detailed explanations and insights, as well as highlighting limitations through the analysis of failure cases.

8. In response to reviewer suggestions, we have included applications, limitations, and future directions in our conclusion.

9. To address concerns about reproducibility, open sourcing, and data release, we have incorporated a reproducibility statement in our manuscript.

We hope these revisions address your concerns and improve the quality of our submission. Once again, we appreciate your constructive feedback and the opportunity to enhance our work.

---

### Meta-Review · Area_Chair_Yn4R · 2023-12-07

**Metareview:**

The reviewers found the approach of utilizing gaze in VLMs interesting. For example, Reviewer tDrZ mentioned "The idea of including gaze information to vision-language model is quite interesting, which might be one important aspect when people use the vision-language model in VR/AR scenarios." However, the reviewers notice that the technical novelty of the work is limited that is confounded with the clarity issue with the presentation. Here are some comments during the discussion between the reviewers:

Reviewer gfYi: "After careful consideration, my conviction in favor of Kosmos-2 and Otter remains strong. However, I find the responses regarding technical novelty lacking, especially they emphasize the engineering aspect. While I appreciate the acknowledgment that this is an engineering endeavor, my concern lies in the insufficient experiments to ascertain whether the proposed pipeline will generalize effectively. Furthermore, my concerns about the dataset continue. The incremental improvement in the method's presentation is noted, but there is still room for enhancement. The term "temporarily removing" in relation to interpretability is unclear, and I believe it should not be highlighted as a significant contribution at all. In light of these considerations, my overall assessment remains unchanged." While Reviewer 2Muy is more positive towards the paper, Reviewer 2Muy also pointed out the limited novelty.

All in all, there lacks a strong voice among the reviewers to champion the paper.

**Justification For Why Not Higher Score:**

The paper has limited novelty and unclear presentation.

**Justification For Why Not Lower Score:**

Introducing gaze into VLM is potentially interesting.

---

### Decision · Program_Chairs · 2024-01-16

Reject